# Can Special Light Glasses Reduce Sleepiness and Improve Sleep of Nightshift Workers? A Placebo-Controlled Explorative Field Study

**Mariëlle P. J. Aarts** [1,*], **Steffen L. Hartmeyer** [2], **Kars Morsink** [1], **Helianthe S. M. Kort** [1,3] **and Yvonne A. W. de Kort** [2]

[1] Department of the Built Environment, Eindhoven University of Technology, P.O. Box 513, 5600 MB Eindhoven, The Netherlands; K.Morsink@tue.nl (K.M.); H.S.M.Kort@tue.nl (H.S.M.K.)

[2] Department of Industrial Engineering and Innovation Sciences, Eindhoven University of Technology, P.O. Box 513, 5600 MB Eindhoven, The Netherlands; S.L.Hartmeyer@student.tue.nl (S.L.H.); Y.A.W.d.Kort@tue.nl (Y.A.W.d.K.)

[3] Technology for Healthcare Innovations, Utrecht University of Applied Science, P.O. Box 12011, 3501 AA Utrecht, The Netherlands

\* Correspondence: M.P.J.Aarts@tue.nl; Tel.: +31-402-472-900

**Abstract:** Nightshift workers go against the natural sleep–wake rhythm. Light can shift the circadian clock but can also induce acute alertness. This placebo-controlled exploratory field study examined the effectiveness of light glasses to improve alertness while reducing the sleep complaints of hospital nurses working nightshifts. In a crossover within-subjects design, 23 nurses participated, using treatment glasses and placebo glasses. Sleepiness and sleep parameters were measured. A linear mixed model analysis on sleepiness revealed no significant main effect of the light intervention. An interaction effect was found indicating that under the placebo condition, sleepiness was significantly higher on the first nightshift than on the last night, while under the treatment condition, sleepiness remained stable across nightshift sessions. Sleepiness during the commute home also showed a significant interaction effect, demonstrating that after the first nightshift, driver sleepiness was higher for placebo than for treatment. Subjective sleep quality showed a negative main effect of treatment vs. placebo, particularly after the first nightshift. In retrospect, both types of light glasses were self-rated as effective. The use of light glasses during the nightshift may help to reduce driver sleepiness during the commute home, which is relevant, as all participants drove home by car or (motor) bike.

**Keywords:** shift work; rapidly rotating; short-wavelength light; sleep; alertness; care professionals

---

## 1. Introduction

Shift work is a common working condition, with over 18% of employees in the EU working according to a shift schedule and over 13% working occasionally at night [1]. Shift work, especially when involving work at night, has been found to affect sleep [2–4], health, well-being [5–9], and performance at work [10,11]. Although the underlying mechanisms are not fully understood, it is thought that the complex interaction of circadian disruption, disturbed sleep, risk behaviors, and psychosocial stress increases the risk of adverse health effects and performance impairments [6,12–14].

Since light is the main synchronizer of the human circadian system, careful manipulation of the light exposure pattern can be used to align the circadian rhythm with the shift work schedule [15]. Although several studies have demonstrated the effectiveness of light during nightshift work, a recent systematic review concluded that the evidence supporting a positive effect of controlled light exposure is still too weak to draw definite conclusions [16]. A detailed analysis was conducted of the field

studies identified in two earlier systematic review articles [16,17]. Of the 17 eligible field studies, the applied light protocols, the outcome measures, and the results were examined. In Table A2, Appendix A, the results are given. Despite the large methodological diversity, most studies found significant effects on at least one of the pre-defined outcome measures. This methodological diversity (in e.g., outcome measures, shiftwork schedules, and the applied treatment protocols), the lack of detail in the description of the light stimulus and procedure, along with other methodological issues (i.e., the absence of a control or placebo condition and the absence of continuously measured corneal light exposure in the appropriate units and quantities) makes it impossible to infer an appropriate light schedule for nightshift workers.

Additionally, all reviewed field studies applied static, location-bound lighting solutions. The suitability of such installations is debatable in organizations where people, such as hospital nurses, follow rapidly rotating shift cycles. As full circadian adaptation is not desirable in rapidly rotating shifts, scheduled light exposure should account for the number of consecutive days of nightshift work and the timing in this cycle for each individual shift worker. Therefore, nurses should receive a personalized light protocol instead of being exposed to the same ambient light protocol as the entire ward. The potential burden and intrusiveness of a light treatment are serious factors to be dealt with when considering the application in the field. A wearable device, such as light glasses, can be particularly useful for shift workers in extremely dynamic working environments, such as hospitals, due to their flexibility and the possible application of individual intervention protocols. However, light glasses, as such, have not yet been studied as an intervention for shift workers.

The aim of this study was to examine whether a novel intervention using light glasses can reduce sleepiness and sleep complaints and improve the recovery of hospital nurses working rapidly rotating (night) shifts. A partial entrainment protocol was applied in order to avoid difficulties re-entraining after the short nightshift cycles of rapidly rotating shift workers. This should generate a compromise phase position for daytime as for nighttime sleep [15]. Earlier studies reported that partial entrainment positively affects alertness, mood, and cognitive performance, as well as daytime sleep length [18–20] under simulated nightwork conditions. However, to our knowledge, no field study has reported on the effectiveness of a partial entrainment protocol for reducing sleepiness and supporting the recovery after nightshifts.

The hypothesis was that when wearing the light glasses, participants would be more alert during wake-time—i.e., both during nightshift time, as well as during leisure time after the nightshift. When wearing the glasses, we also expected improvements in daytime sleep after the nightshift (and a faster recovery process after the nightshift period).

## 2. Results

Several indicators for sleep, sleepiness, and alertness were continuously monitored in a field study.

### 2.1. Sleepiness (Karolinska Sleepiness Scale)

Figure 1 shows the estimated marginal means (EMM) of the Karolinska Sleepiness Scale (KSS) for Condition (treatment, placebo), Daytype (first (D1) and second (D2) day before nightshift), first nightshift (FN), nightshift (N), last nightshift (LN), first (R1) and second (R2) recovery day, and Time of day (two-hour interval) during times awake. During nightshift work, the linear mixed-effects models (LMM) analysis (see Appendix B Table A3) revealed no significant main effect of the Condition. However, the interaction Condition × Daytype showed significant differences. Under the placebo condition, sleepiness was significantly higher on the FN ($1.58 \pm 0.06$) compared to the LN ($1.47 \pm 0.05$; $p = 0.002$), while sleepiness under the treatment condition remained stable across Daytype, FN ($1.53 \pm 0.06$), compared to LN ($1.52 \pm 0.05$; $p = 1$). The $ICC_{Subject}$ was 0.39. The interaction effect between Condition × Daytype × Time of day showed a significant difference between the placebo and treatment on the FN at 4 h, on N at 8 h, and on R1 at 18 h. For the other periods, no significant main effects were found

for Condition, nor the interaction Condition × Daytype. Significant main effects for all periods were found for Daytype and Time of day.

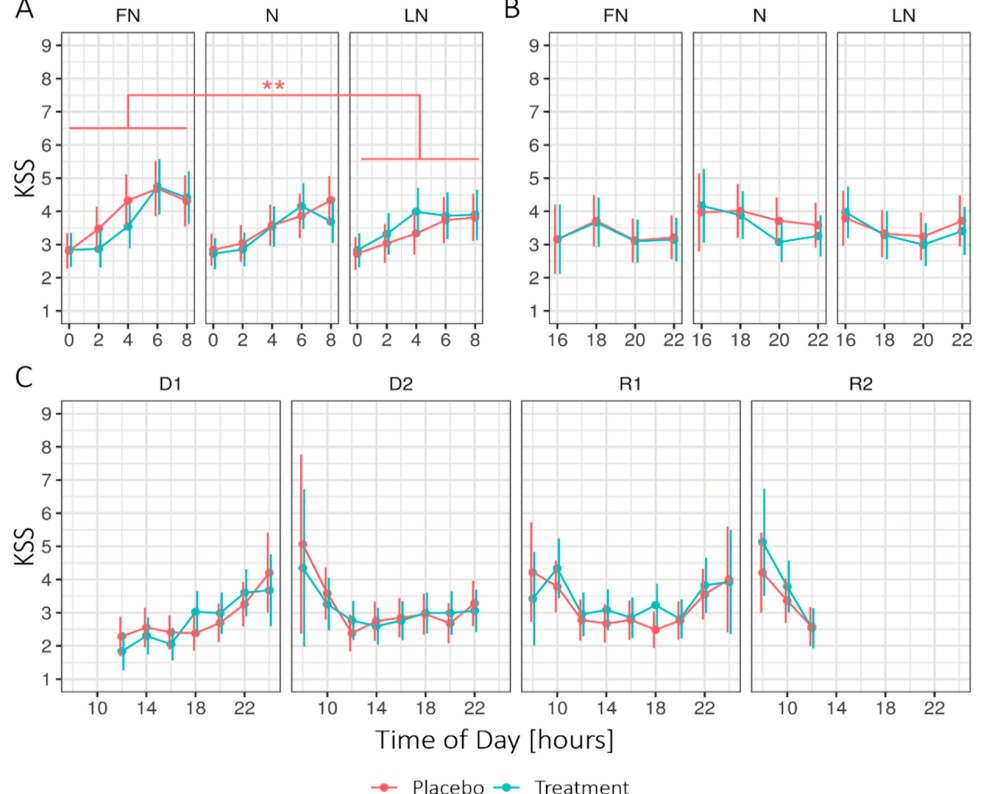

**Figure 1.** Estimated marginal means for the Karolinska Sleepiness Scale (KSS) in 2-h intervals, (**A**) during nightshift work, (**B**) after nightshift work, and (**C**) on the first two days and recovery days for the treatment and placebo condition. Higher values indicate higher levels of sleepiness. Error bars indicate the 95% confidence intervals of the estimated marginal means. D1 = first day before nightshift, D2 = second day before nightshift, FN = first night of nightshift, *N* = nightshift, LN = last night of nightshift, R1 = first recovery day, and R2 = second recovery day. For all time periods, Conditions did not significantly differ per day. Note that significant effects at the level of Time of day are not displayed. ** = $p < 0.01$.

## 2.2. Driver Sleepiness (Driver Sleepiness Scale)

Figure 2 shows the EMMs for the driver sleepiness scale (DSS) and KSS when returning home after nightshift days for both experimental conditions. The LMM analysis (See Appendix B Table A4) of the DSS revealed no significant main effect of Condition; however, the interaction Condition × Daytype was significant ($F_{2130.4} = 3.60$, $p = 0.030$, $R^2 = 0.05$). The interaction Condition × Daytype entailed that, during the first nightshift (FN) but not the subsequent days, the DSS was higher for the placebo ($3.30 \pm 0.49$) than for the treatment ($2.17 \pm 0.49$; $p = 0.012$). Moreover, for the placebo, the DSS was significantly lower on the LN ($1.22 \pm 0.50$) compared to the FN ($3.30 \pm 0.49$; $p < 0.001$) and N ($2.69 \pm 0.47$, $p < 0.01$), whereas the DSS stayed stable under the treatment condition. A significant main effect for Daytype was found. For the KSS, no significant main or interaction effects were found. Four participants reported eight near-accidents in total on their commute home, equally under treatment (4) and placebo (4).

## 2.3. Sleep

Figure 3 shows the EMMs of the different sleep parameters across Condition (placebo, treatment) and Daytype (D2, FN, N, LN, R1, R2). The LMM analysis (see Appendix B Table A5) revealed

a significant main effect of Condition for the Groningen Sleep Quality Scale (GSQS; the GSQS under treatment was higher—i.e., worse—than under placebo). The interaction Condition × Daytype was significant for time in bed (TIB), total sleep time (TST), and sleep latency (SL).

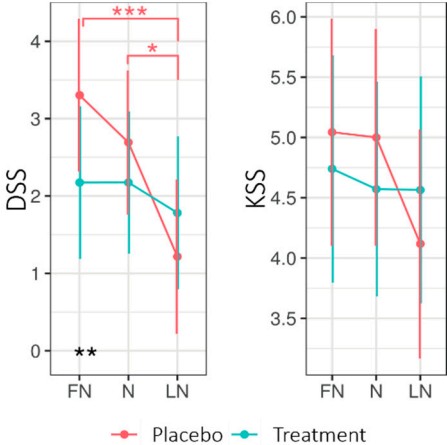

**Figure 2.** Estimated marginal means for Driver Sleepiness (DSS) and Sleepiness (KSS) for the commute home for the treatment and placebo condition across nightshift days. Higher values indicate higher levels of sleepiness on both scales. Error bars indicate the 95% confidence intervals of the estimated marginal means. * = $p < 0.05$; ** = $p < 0.01$, *** = $p < 0.001$. FN = first night of nightshift, N = nightshift, LN = last night of nightshift.

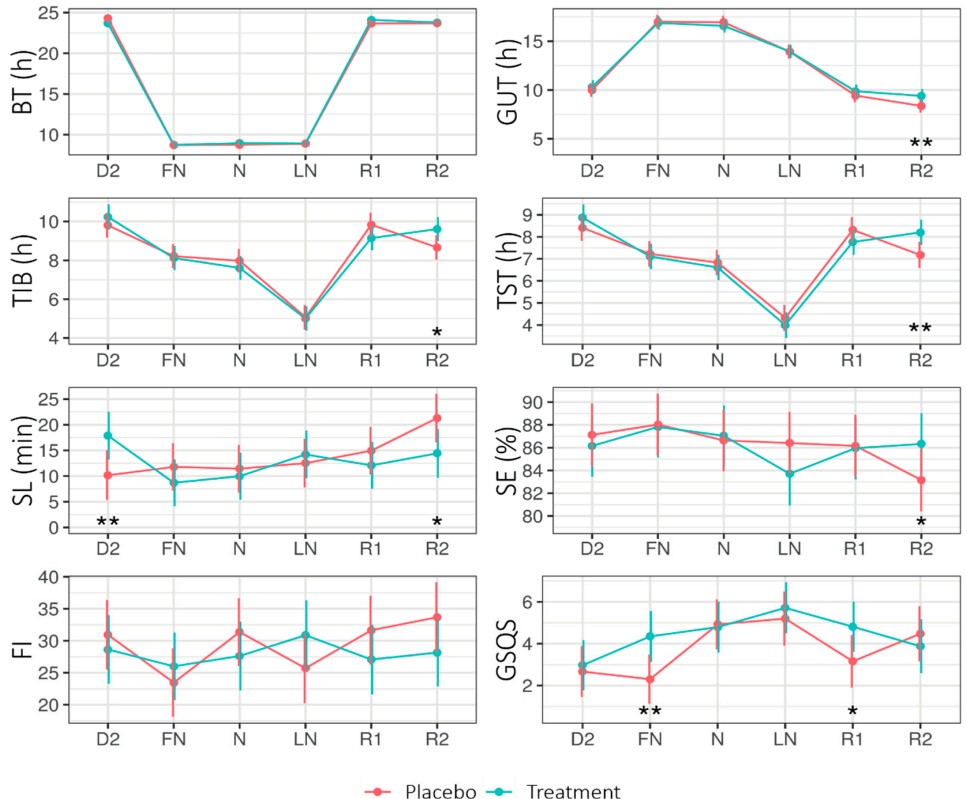

**Figure 3.** Estimated marginal means (EMM) of bed time (BT), get-up time (GUT), time in bed (TIB), total sleep time (TST), sleep latency (SL), sleep efficiency (SE), fragmentation index (FI), and Groningen Sleep Quality Scale (GSQS) score across Daytype for the placebo and treatment condition. Error bars indicate the 95% confidence intervals of the estimated marginal means. D2 = night before nightshift, FN = first night of nightshift, N = nightshift, LN = last night of nightshift, R1 = first recovery day, and R2 = second recovery day. * = $p < 0.05$, ** = $p < 0.01$.

On the second recovery day (R2), SL was shorter for treatment (14.42 ± 2.37) than placebo (21.31 ± 2.37; *p* = 0.020), while on the second day (D2: i.e., before the first nightshift and light intervention), SL was longer for treatment (17.88 ± 2.32) than placebo (10.17 ± SE = 2.42; *p* = 0.01). On R2, the TIB and TST were longer for the treatment than for the placebo.

### 2.4. Person-Bound Light Exposure

Figure 4 shows the mean illuminance per hour across Daytype. For the periods of nightshift work and the morning commute home, the LMM analyses revealed no significant main or interaction effects for Condition or Daytype. For the period of nightshift work and the morning commute home, no significant main or interaction effects on the exposure to high light levels (i.e., minutes of illuminance >1000 lux) were found.

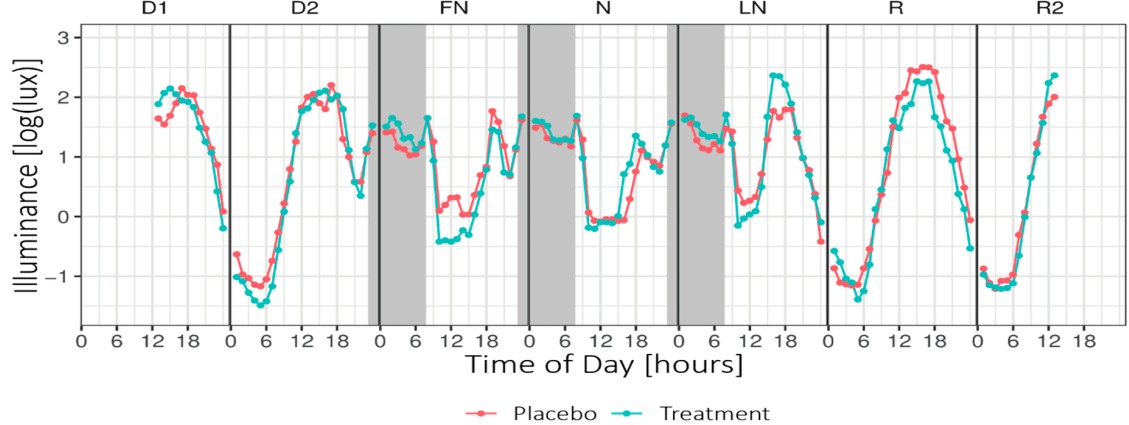

**Figure 4.** Mean hourly illuminance (log-transformed) across Daytype. Grey areas indicate nightshift work (~23:15–07:45). Note that the data presented in this plot were not adjusted for the transmission of the blue-blocking goggles and do not include light exposure from the light glasses. D1 = first day before nightshift, D2 = second day before nightshift, FN = first night of nightshift, N = nightshift, LN = last night of nightshift, R = first recovery day, and R2 = second recovery day.

### 2.5. Participants' Subjective Experiences of Both Light Glasses in Retrospect

The results of the participants' retrospective results are presented in Table 1. Although positive effects were more frequently reported for the treatment light glasses, for all measures, at least 20% of the participants indicated positive effects of the placebo glasses as well. Only for alertness was there a significantly higher average rating for the treatment glasses than for the placebo glasses (4.17 vs. 4.83, with 7 = much more alert; *p* = 0.015). Seventeen of the participants indicated that they would wear the treatment glasses in the future versus seven who would wear the placebo version, despite comfort issues. Only one participant indicated an intolerable interference of the light glasses when working [21].

**Table 1.** Assessment of participants' retrospective results of both light glasses (*N* = 23).

|  | Placebo | | | Treatment | | |
|---|---|---|---|---|---|---|
|  | Mean ± SD | Number of Participants Rating [-] | | Mean ± SD | Number of Participants Rating [-] | |
| Effect on |  | Positive | Negative |  | Positive | Negative |
| Alertness [a] | 4.17 ± 0.58 | 4 | 1 | 4.83 ± 1.03 | 11 | - |
| Sleep quality [a] | 4.26 ± 0.62 | 4 | - | 4.48 ± 0.99 | 8 | 1 |
| Sleep length [a] | 4.22 ± 0.52 | 4 | - | 4.13 ± 0.87 | 5 | 3 |
| Recovery [a] | 4.48 ± 0.67 | 9 | - | 4.52 ± 0.99 | 9 | 1 |

**Table 1.** *Cont.*

| | Placebo | | Treatment | |
|---|---|---|---|---|
| | **Mean ± SD** | **Number of Participants Rating [-]** | **Mean ± SD** | **Number of Participants Rating [-]** |
| Wear [b] | 4.20 ± 1.15 | | 4.40 ± 1.14 | |
| Comfort [b] | 3.90 ± 1.29 | | 3.90 ± 1.29 | |
| Adverse effects | | | | |
| Headache | | 6 | | 4 |
| Impaired vision | | 3 | | 5 |
| Dizziness | | 2 | | 0 |
| Concentration | | 1 | | 1 |

[a] On a scale from 1–7, with 1 = much worse, 4 = no effect, 7 = much better; [b] On a scale from 1–7, with 1 = very unpleasant/uncomfortable, 7 = very pleasant/comfortable.

## 3. Discussion

This exploratory placebo-controlled field study investigated the impact of light glasses ($\lambda = 462 \pm 10$ nm, spectral irradiance of 22.36 µW/cm$^2$ at the cornea of the eyes) administered according to a partial entrainment protocol to reduce sleepiness during and after the nightshift without negatively impacting the sleep of hospital nurses working rotating nightshifts. Circadian adaptation and a qualitative good sleep contribute to preventing long-term adverse health effects of (night) shiftwork [6,12].

### 3.1. Effects of the Light Treatment

#### 3.1.1. Driver Sleepiness

With the treatment glasses, the driver sleepiness on the commute home after the first nightshift was significantly lower. While the driver sleepiness reduced over the different nightshift days (FN, N, and LN) under placebo conditions, it remained stable over the treatment period. The lower driver sleepiness under treatment conditions after the first nightshift suggested a potential risk reduction due to the light glasses for participants on their commute home, although we should note that this effect did not persist on subsequent days. Although drowsiness on the commute home is acknowledged as a major problem, light is seldom applied as a prevention strategy [22]. These findings emphasize the relevance of addressing the safety of care professionals after night work on the commute home, especially considering the high number (four under treatment and four under placebo) of reported near-accidents in this study.

#### 3.1.2. Sleepiness (KSS)

Under placebo conditions, the sleepiness on the first nightshift was significantly higher than on the last nightshift. Regarding treatment conditions, no significant difference was found over the three nightshift periods. This indicates that the sleepiness remained the same during the nightshift period. More specifically, significant differences between placebo and treatment on the FN at 4 h, on the N at 8 h, and on the R1 at 18 h were found. During the nightshift, under treatment conditions, participants were less sleepy than under placebo conditions and on the first recovery day, they were sleepier, yet, no main effect for treatment on sleepiness was found, while other studies, using scheduled light exposure during the nightshift, did report such effects (see Appendix A Table A2). Perhaps the (scheduled) light exposure after waking up counteracted the phase delaying effects of light during the night, as Smith et al. [19] suggest. Alternatively, the increased sleepiness on the first recovery day might be explained by the phase delay due to the nightly light exposure, but this was not supported by the absence of effects during the last nightshift. Unfortunately, we were unable to test either explanation as we did not include circadian markers.

For reference, the only other comparable shift work study that used narrowband short-wavelength light [23] did observe significant improvements in sleep parameters, subjective alertness, and work errors, with a more intense light stimulus (~66 $\mu W/cm^2$ at 500 nm vs. ~23 $\mu W/cm^2$ at 461 nm in the present study, on top of the ambient light conditions). The treatment period between three to five consecutive nights in our study may have been too short to produce measurable effects. Previous research reported substantial improvements after five consecutive nights: e.g., [19,24,25]. In the present study, only one participant worked a period of five consecutive nights.

### 3.1.3. Sleep

A treatment effect could be observed for time in bed (TIB), sleep duration (TST) and sleep latency (SL), get-up time (GUT), and sleep efficiency (SE) on the second recovery day (R2). However, further analyses showed that the significant effects on these sleep outcomes were largely influenced by six participants who were scheduled to work a morning shift (starting at 07:30) on the second recovery day. After excluding the data of these six participants, the impact on TIB, TST, GUT, SE, and SL on the last recovery day and BT on the second day were no longer significant at a 0.05 level. Nevertheless, even after excluding these data from the analysis, sleep duration remained longer (48 min) under the treatment condition ($p = 0.094$). These results are consistent with previous studies that found positive effects of light exposure on sleep duration and sleep efficiency after waking up on days off after a period of night work [26], as well as sleep latency [27]. Positive treatment effects on subjective sleep quality [3,26,28] could not be confirmed in the present study. In contrast, the sleep quality (GSQS) in our study was significantly worse under treatment than placebo conditions, specifically after the first nightshift and on the first recovery day. This finding would be in line with the amplification of after-effects due to light-induced circadian adaptation [29] and deserves more attention in future studies.

### 3.2. Light Exposure

Expressed in illuminance values, the light glasses provided an ocular exposure of 15.54 lx. The ambient illuminance during the nightshift was, on average, 84 lx. This means that for 1 h of treatment, an additional 20% of photopic light was offered. The addition of light when wearing the light glasses for 30 min after daytime sleep was negligible (1%) compared to the normal ambient light conditions. For a fair comparison, the irradiance measured from the light glasses, measured at cornea position by the blue sensor (peak 465 nm) of the LightWatcher of the glasses gave a value of ~0.6 $mW/m^2/nm$. During the nightshift, the average ambient irradiance for that specific sensor was 0.8 $mW/m^2/nm$ (treatment) and 0.7 $mW/m^2/nm$ (placebo). This means that the irradiance (around 465 nm) from the ambient lighting of the hospital environment had the same order of magnitude as the light glasses—so a total of ~1.4 $mW/m^2/nm$ in the bluish part of the spectrum (around 460–465 nm). In a recent publication, Brown [30], based on 19 studies, concluded that melanopic illuminance, expressed in melanopic equivalent daylight illuminance (EDI lux), is the best available predictor for responses of the human circadian system. His analyses show that the melanopic EDI 114 lx for 1 h, as provided by the light glasses, is capable of impacting melatonin suppression, circadian effects, and alerting responses.

### 3.3. Limitations and Future Studies

Given the current study design, several limitations must be considered. First, sleepiness was sampled every two hours on a subjective scale (KSS). Additional objective correlates of performance could have given more insight into the existence and strength of placebo effects. A recently conducted field study [31] monitored the alertness of shift-working intensive care workers subjectively by the KSS and by performance on the Psychomotor Vigilance Test (PVT). Both outcome measures showed a similar trend, implying that KSS, as used in our study, is a relatively good measure for indicating the sleepiness of nightshift workers. Sampling circadian phase markers could have served as a manipulation check and given more insight into whether the light treatment delayed the circadian rhythm. On the

other hand, additional measurements would have interrupted the participants' working routine. Additionally, the use of KSS, a one item-scale, might not be sensitive enough for indicating sleepiness. While the driver sleepiness scale (DSS), a multiple-item scale, revealed significant effects for the commute home, the KSS, measured at the same time, did not, although both scales have been shown to correlate. The driver sleepiness scale (DSS) might be a more sensitive indicator of sleepiness than the KSS and, therefore, may be more sensitive to a potential phase delay due to nocturnal blue light exposure, especially when homeostatic sleep pressure is high. In future studies, it may be useful to increase the sampling frequency of the KSS to every hour and include a possibility for participants to indicate spontaneous variations in alertness: for example, when an emergency occurs.

Second, sleep parameters were measured using actigraphy. The accuracy of actigraphy is limited without the additional use of sleep logs [32]. In the present study, the bed and wake times used in the actigraphy analysis were obtained from a combination of the subjective sleep logs and timings inferred from the light logger and actigraphy data. However, the discrepancy between the subjective and objective timings was often high. A solution for future studies may be the use of an event marker that is activated at lights-out.

Third, light exposure was measured using six irradiance sensitive sensors (see supplementary material Figure S1 for the spectral sensitivity of the sensors of the LightWatcher). The picture in Figure 5 shows the spectral power distribution of the light glasses (on the cornea) and the action spectrum for melanopic irradiance. This indicates that the light emission by the light glasses had a peak at 461 nm, while the action spectrum for melanopic irradiance peaks at 480 nm. Therefore, it could be that both spectra were not aligned, and the results might have been bigger when the peak of the ocular irradiance was closer to 480 nm. Additionally, the LightWatchers were worn at chest height. Median deviations of 17% for indoor conditions and 7% for daytime outdoor conditions were reported between the illuminance measurements of the chest and eye position [33].

Fourth, the study was partly conducted in the brighter part of the year (last participants in August). The high daylight levels might have impacted the outcomes. A future study in winter is essential to indicate the robustness of the results and it is expected that the impact might even be stronger given lower light exposure history during daytime [34].

Fifth, inter-individual variation is a concern that has been frequently reported for light exposure interventions (e.g., Kantermann et al. [35]). Specifically, individuals often differ in their tolerance to shift work [36] and circadian characteristics [37], which makes it difficult to develop general intervention protocols. The results of the present study showed that for most outcome measures, a substantial proportion of variance was accounted for by inter-individual differences. This finding highlights the strong and consistent impact of sleeping at the wrong circadian time on sleep length, whereas for other sleep parameters, differences between daytime and nighttime sleep may be more nuanced.

Other limitations and potentially confounding factors are individual coping strategies (e.g., power naps, caffeine intake, food intake, etc.), the fact that one of the treatment light glasses emitted significantly less light, as well as potential administration deficiencies of the light glasses.

## 4. Materials and Methods

### 4.1. Study Design

This study employed a placebo-controlled single-blinded cross-over within-subjects design, comprising two experimental conditions: one treatment condition (treatment) and one placebo condition (placebo). The study was carried out between February and August 2019 within the participants' regular work schedule. The measurement equipment, together with an individual verbal and written instruction, was personally handed to the participants before the start of the measurements. An overview of the study procedure is depicted in Figure 6. The participants were instructed to keep a study diary to indicate the time periods that the light glasses were applied, the measurement devices were not worn,

the consumption of stimulating drinks and substances, and commuting to and from work. Furthermore, participants were requested to refrain from using an alarm to wake up and to avoid napping.

## 4.2. Participants

The participants were recruited among nurses working rotating nightshifts in direct patient care at a University Medical Center in the Netherlands. Via in-service presentations, flyers, and e-mail contact, 28 nurses agreed to participate in this study. The inclusion criterion was that they worked two rows of 3–5 consecutive nightshifts within the study period. The exclusion criteria were: having worked nightshifts within a week before the measurement period, wearing corrective glasses, extreme morning or evening chronotypes, pregnancy, a diagnosed sleep or psychological disorder, eye-related diseases, or a recent stay in a location more than three time zones away.

Out of the 28 initial participants, five could not complete both measurement periods. Headache, presumably from wearing the light glasses, was the only study-related reason ($n = 1$). The remaining 23 participants (20 female) were included in the final data analysis (see Table 2) for the demographic data. The participants worked on various wards of the same hospital according to a three-shift schedule of 8 h per shift. The nightshift started between 23:00 h–23:30 h and ended between 7:30 h–8:00 h. Nine participants worked three nightshifts in both periods, three worked four shifts in both periods, nine worked three and four shifts, one worked three and five shifts, and one worked four and five shifts. The working schedules were not fixed for 84% of the participants. Although a forward-rotating schedule was recommended, only one third followed such a schedule. The chronotype was assessed using the Munich Chronotype Questionnaire for Shift Workers (MCTQ Shift [38]). The chronotype could not be determined for eleven participants since they always used an alarm clock to wake up. The study protocol was approved by the ethical committee of the Human-Technology Interaction group (Eindhoven University of Technology, reference number ID 904). The Medical Ethics Review Committee of RadboudUMC (Arnhem-Nijmegen region) considered the study not subject to the Act of Medical Research Involving Human Subjects (ref. 2018-4965). Each participant signed the approved consent form and received a voucher upon the completion of both measurement periods.

**Table 2.** Demographics of participants ($N = 23$).

|  | **M** | **SD** | ***n*** |
|---|---|---|---|
| **Age (years old)** | 30.1 | 10.2 | |
| **Working experience in healthcare (years)** | 9.13 | 11.5 | |
| **Working week (hours)** | 33.6 | 3.03 | |
| **Monthly number nightshifts (-)** | 5.66 | 2.38 | |
| **Chronotype (hh:mm)** | 04:40 | 01:07 | |
| **Late (MSF$^E_{SC}$ [b] > 5)** | | | 4 |
| **Intermediate (MSF$^E_{SC}$ = 03:00–05:00)** | | | 7 |
| **Early (MSF$^E_{SC}$ < 3)** | | | 1 |
| **Undefined** | | | 11 |

[b] MSF$^E_{SC}$ = Mid-sleep phase corrected for shift workers [38].

## 4.3. Light Protocol

The light protocol was based on Smith and colleagues [19]. Commercially available light therapy glasses (Propeaq Premium Light glasses, Chrono Eyeware B.V., Tilburg, The Netherlands) were used. For the treatment condition, the commercially available version was used containing integrated LEDs in the frame ($\lambda = 461 \pm 10$ nm, spectral irradiance of 22.34 $\mu$W/cm$^2$/melanopic EDI of 114.38 lx at the cornea of the eyes). In the placebo condition, the LEDs of the otherwise identical version of the glasses used in the treatment condition shone on the temples, resulting in no measurable corneal light exposure (see Figure 5 for images and detailed light characteristics).

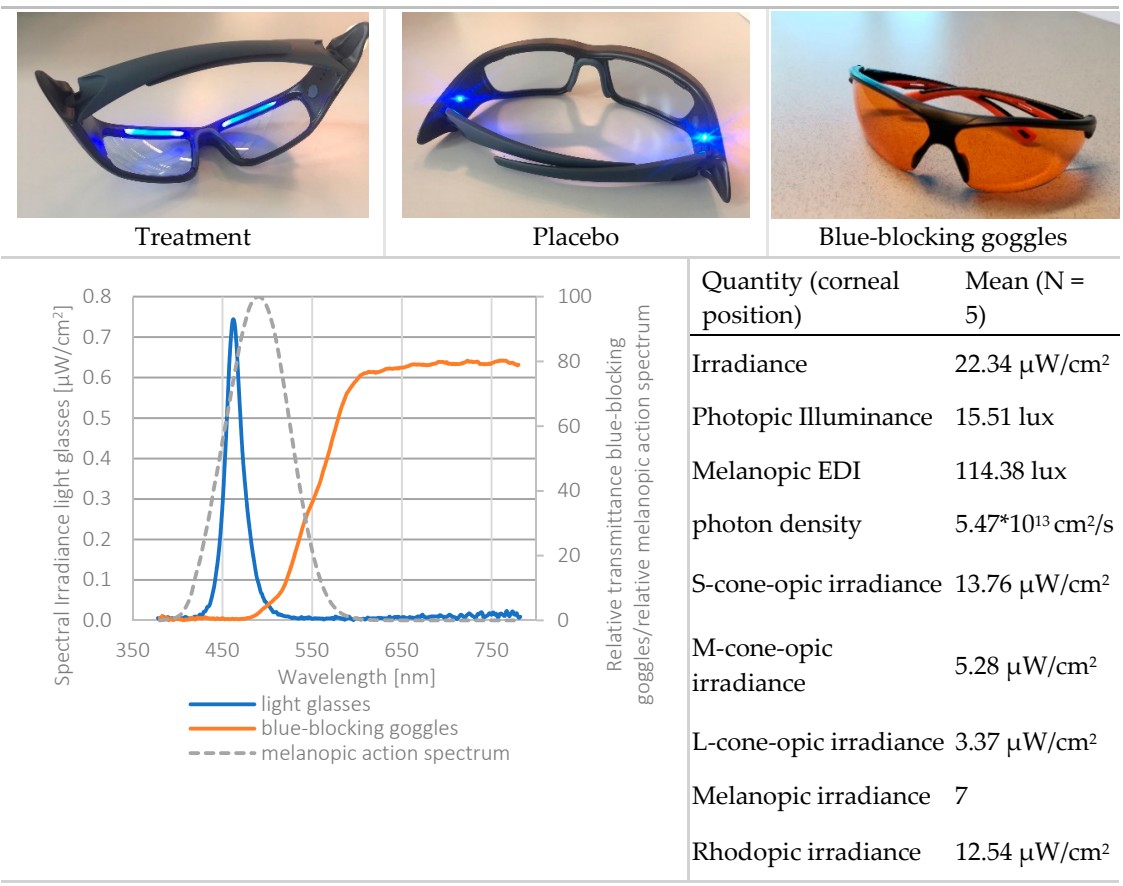

**Figure 5.** Top: light glasses (treatment, placebo) and the blue-blocking goggles. Bottom left: spectral power distribution of the treatment light glasses, melanopic irradiance, and the spectral transmission of the blue-blocking goggles. Light measurements were performed at corneal position, using a realistic head model. Bottom right: radiometric and photometric properties of the treatment light glasses. Calculations through the CIE S 026 Toolbox—beta version E1.05 [39].

In both conditions, light exposure (blue areas indicating times when light glasses were worn, in Figure 6) was administered 4 × 15 min during nightshifts and for 30 min within 2 h after awakening. Participants were asked to wear orange-tinted blue-blocking goggles during the morning commute home in order to ensure relatively equal morning (blue) light exposure within and across participants over the entire study period (see Figure 5, bottom left for the relative spectral transmittance distribution). Four participants used their own sunglasses for reasons of safety while driving. The sunglasses' luminous transmittance ranged between 11–30% and 0–5% in the visible and short wavelength range, respectively.

Study Design

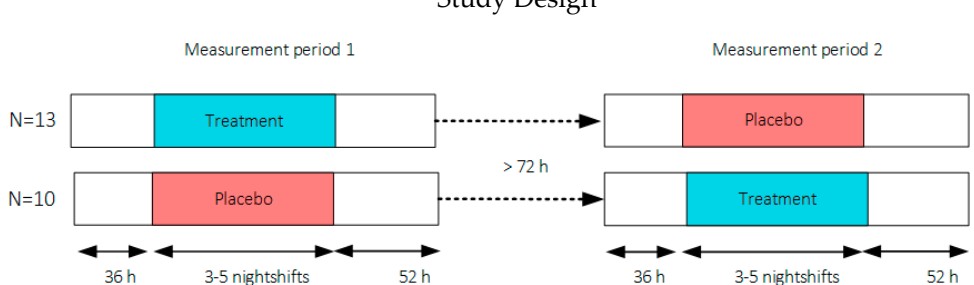

Light exposure protocol and measurements.

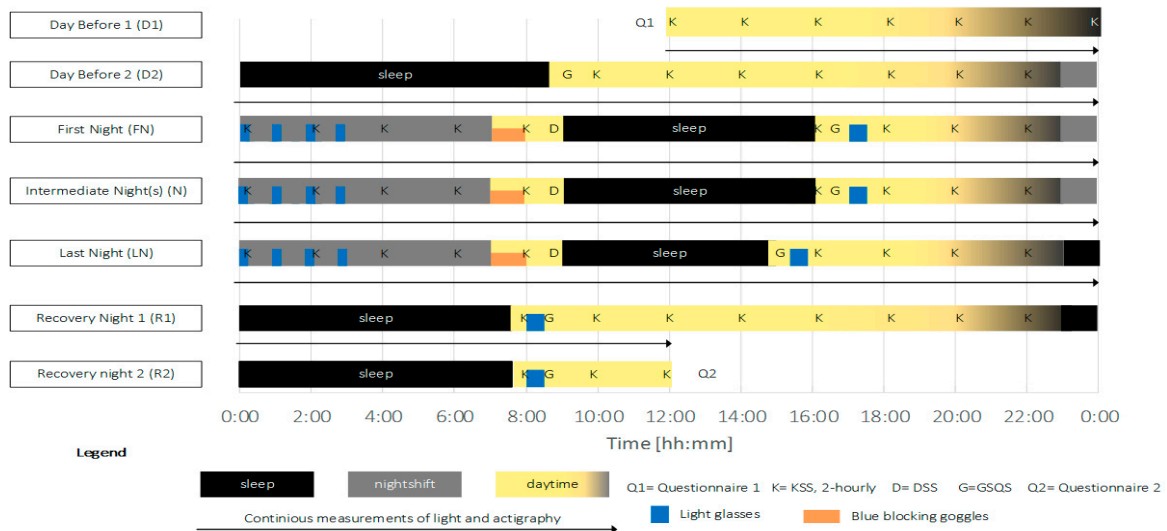

**Figure 6.** Top: overview of the study design, Bottom: exemplary light exposure protocol and measurement timings for a three-day measurement period. Participants started at 12:00 the day before (D1) the first nightshift, slept (black areas, D2) as normal and started their first nightshift (gray areas, end of D2). Light exposure (blue areas) was administered 4 × 15 min during nightshifts (FN, N, and LN) and for 30 min within 2 h after awakening. During the morning commute home, blue-blocking goggles were worn (orange areas). The measurement period ended at 12:00 two days after the last nightshift (R2). KSS (K) was measured every two hours during the wake period. The driver sleepiness questionnaire (D) was filled in before going to bed after nightshifts. The GSQS (G) and sleep diary were filled in directly after sleep. Objective sleep and person-bound light exposure were continuously monitored during the entire measurement period.

## 4.4. Measurements

### 4.4.1. Sleepiness (KSS)

Sleepiness was assessed using the KSS (Karolinska Sleepiness Scale [40]) during wake times via the PRO-Diary (PRO-Diary, CamNtech Ltd, Papworth Everard, UK). The PRO-Diary is a compact electronic diary, which, for reasons of hygiene according to hospital regulations, was worn on the upper arm when on duty, and on the non-dominant wrist outside work hours. Every two-hours, participants were prompted to indicate their sleepiness on a 9-point Likert scale ranging from 1 (extremely alert) to 9 (very sleepy, fighting sleep). The timing was individually programmed based on the expected wake times indicated by the participant.

### 4.4.2. Driver Sleepiness (DSS)

Sleepiness during the morning commute home was measured using a scale adopted from Watling, Armstrong, Smith, and Wilson [41], which was self-administered via the PRO-Diary mentioned in previous paragraph when arriving home after the nightshift. The driver sleepiness scale (DSS) comprises signs of sleepiness while driving based on eight items. Additionally, the participants were asked to indicate their sleepiness (KSS) during the commute home and whether accidents or near misses had occurred.

### 4.4.3. Objective Sleep Measures (Actigraphy)

Sleep–wake parameters were assessed by actigraphy using the PRO-Diary. Wrist activity data were analyzed using Actiwatch Sleep Analysis software (Version 5.11; CamNtech Ltd.). Sleep periods were determined from self-reported bedtimes, illuminance data (i.e., on- and offset of darkness), and activity data. In the case of diverging indications of the self-reported bedtimes and the illuminance and activity data, bedtimes were taken from the illuminance and activity data. The derived sleep variables were bed time (BT), sleep latency (SL), get-up time (GUT), time in bed (TIB), total sleep time (TST), sleep efficiency (SE; TST divided by TIB), and a sleep fragmentation index (FI = mobile time (%) + immobile bouts ≤ 1 min (%)).

### 4.4.4. Subjective Sleep Measures (GSQS)

Self-reported sleep quality was assessed with the Groningen Sleep Quality Scale (GSQS by Mulder-Hajonides Van Der Meulen and colleagues [42]), which was self-administered via the PRO-Diary after every sleep episode. The GSQS assesses sleep quality on a 14-item scale, with scores ranging from 0–14. Scores between 0 and 2 indicating normal sleep, and scores ≥ 6 indicated disturbed sleep [43].

### 4.4.5. Person-Bound Light Exposure

To confirm the absence of a confounding factor between both treatments, the person-bound ambient light exposure was continuously recorded. A calibrated wearable light logging device (LightWatcher, Wolf Technologie—Object Tracker, Perchtoldsdorf, Austria), was used, recording photopic illuminance and spectral irradiance at five different wavelengths. The participants were asked to wear the LightWatcher on their outer layer of clothing at chest height during their wake period. During sleep, participants were instructed to place the device, sensors facing upward, beside the bed. Under all conditions, the sensors measured in the horizontal plane, facing upward. The light loggers were calibrated according to the procedure described in [33] (see also supplementary material for the calibration procedure). Every 10 s, a measurement was taken and automatically merged into 30-s averages. For the analyses, these values were processed in one-minute averages.

### 4.4.6. Perceived Effectiveness

After the completion of both measurement periods, the participants were asked to answer a list of questions inquiring about the perceived effects of both versions of the glasses. These questions were included in order to check the credibility of the placebo and potential biases.

### *4.5. Data Analysis*

Before the analysis, the PRO-Diary data were inspected for anomalies and discarded if the data suggested that the participants did not wear the device (e.g., no motion activity of actigraphy during the entire sleep period). Statistical analyses were conducted using the open-source statistical computing software R (version 3.5.1). Unless stated otherwise, analyses of the dependent measures were performed using linear mixed-effects models (LMM). The models were fitted with combinations of the independent fixed factors Condition (placebo, treatment), Daytype (D1: first day, D2: second day, FN: first nightshift, N: intermediate nightshifts 3F[1], LN: last nightshift, R1: first recovery day,

R2: second recovery day), and Time of day (hours), and a random intercept for subject. Post-hoc analyses were performed with pairwise Tukey multiple-comparison tests of estimated marginal means (EMM). The proportion of variance accounted for by the grouping structure was given as an interclass correlation coefficient (ICC). All statistical tests were performed using a two-sided significance level of 0.05. The results were expressed as EMM ± standard error (SE) unless stated otherwise.

### 4.5.1. Sleepiness

The responses on the KSS were averaged into 2-h bins and analyzed with the factors Condition, Daytype, and Time of day separately for the following time periods: during the first days and recovery days (D1, D2, R1, R2), during work on nightshift days (FN, N, LN), and after work on nightshift days. In order to test for acute alerting effects of the treatment light goggles, it was examined whether the KSS responses differed between condition and Daytype for the periods within 2 h after awakening on intervention days (FN, N, LN, R1, R2): that is, when the light glasses were scheduled for 30 min. Note that responses were log-transformed due to substantial skewness.

### 4.5.2. Driver Sleepiness

The responses on the DSS were assessed for reliability and validity before the LMM analysis. The factor analysis demonstrated good internal reliability between the different items. This allowed us to aggregate the items into one driver sleepiness score (DS score) as the unweighted sum of item responses. The resulting DS score ranged from 0–8, with higher scores indicating higher sleepiness while driving. The construct validity of the scale was further supported by a positive correlation of the DS score and the KSS for the commute home (Pearson's $R = 0.67$; $p < 0.001$). Then, the aggregated item score and the responses on the KSS for the way home were subjected to LMM analysis with the factors Condition and Daytype.

### 4.5.3. Sleep

The objective sleep parameters obtained from actigraphy and subjective sleep quality on the GSQS were analyzed for the factors Condition and Daytype. Because the sleep patterns of some participants included substantial irregularities (e.g., due to personal or social events), outliers on the standardized conditional residuals (i.e., smaller/greater than 2.5 standard deviations from the mean) were excluded and the LMMs were re-fitted for the data without the outliers. The percentage of excluded data was 1–3% for the different dependent measures.

### 4.5.4. Person-Bound Light Exposure

The person-bound light exposure data from the LightWatcher were analyzed as mean log-transformed ($\log(x)$) illuminance and log-transformed ($\log(x)$) minutes of high light levels (i.e., >1000 lux). The threshold of 1000 lux was chosen to indicate bright light exposure comparable to stimuli used in intervention studies [17] and to indicate potential daylight exposure, since illuminance values of such magnitude are rarely found indoors for electrical light sources [19]. Data indicating that the LightWatcher was not worn were excluded from the analyses.

## 5. Conclusions

This study demonstrated the potential of using light glasses for reducing sleepiness during the first nightshift.

The sleepiness under the placebo condition was significantly higher on the first night of the cycle than on the last night. When wearing the treatment light glasses, no significant difference was noticed over the different days, although at 4 h on the first night and 8 h on the other nights, the participants were less sleepy than wearing the placebo glasses. The driver sleepiness after the first night was significantly lower with the treatment light glasses than with the placebo light glasses. This finding

suggests the potential of reducing traffic accidents for nightshift workers on their commute home. Light is seldom suggested as a prevention strategy against drowsiness on the commute home.

Even though the subjectively assessed daytime sleep quality after the first nightshift was significantly worse when the treatment light glasses were worn, all other, objectively measured sleep parameters were not negatively impacted by the treatment glasses. These findings suggest that sleep as an indicator for recovery is only mildly impacted by the treatment light glasses.

Although the impact of the light glasses on reducing sleepiness during wake times was weaker than initially expected, almost half (11 of the 23) of the participants indicated that they felt more alert when wearing the light glasses. Furthermore, the majority of participants would use the light glasses or the placebo version in the future, despite comfort issues when wearing the glasses. This indicates that the problem of working nightshift work has reached a point where all opportunities that might help are fully embraced, even without scientific evidence. The flexibility of the light glasses motivates further research to evaluate the effectiveness during the nightshift period with a different intervention protocol.

**Supplementary Materials:** The following are available online at http://www.mdpi.com/2624-5175/2/2/18/s1, Description of the calibration protocol of the LightWatcher. Figure S1: Relative spectral sensitivity of the five sensors (colored lines), action spectrum of melanopic irradiance (light blue dotted line). Pink = UV, Blue = 465 nm, Green = 540 nm, Red = 620 nm, and brown = 875 nm. Table S1: Results of the LightWatchers under treatment conditions, Table S2: Results of the LightWatchers under placebo conditions.

**Author Contributions:** Conceptualization, M.P.J.A., S.L.H., and K.M.; methodology, M.P.J.A., S.L.H., Y.A.W.d.K.; validation, S.L.H., K.M., M.P.J.A., and Y.A.W.d.K.; formal analysis, S.L.H. and K.M.; investigation, S.L.H. and K.M.; resources, M.P.J.A., S.L.H., and K.M.; data curation, S.L.H.; writing—original draft preparation, M.P.J.A., S.L.H., and K.M.; writing—review and editing, M.P.J.A., S.L.H., K.M., H.S.M.K., and Y.A.W.d.K. visualization, S.L.H., K.M., and M.P.J.A.; supervision, M.P.J.A., H.S.M.K., and Y.A.W.d.K. All authors have read and agreed to the published version of the manuscript.

**Funding:** This research received no external funding.

**Acknowledgments:** We thank all participants for their dedication and contribution to this study. We also thank the hospital organization for facilitating this study. We acknowledge Marijke Gordijn for her valuable input on the study design. Chrono eyewear is thanked for lending us the light glasses. The Utrecht University is gratefully thanked for lending us the Light Watchers. Finally, we thank Alexander Rosemann of the Building Lighting group (TUE) for his support, and Wout van Bommel and Jan Diepens of the Building Physics and Services lab (TUE) for their technical support.

**Conflicts of Interest:** The authors declare no conflict of interest. No others than the authors had a role in the design of the study; in the collection, analyses, or interpretation of data; in the writing of the manuscript, or in the decision to publish the results.

## Abbreviations

| | |
|---|---|
| KSS | Karolinska Sleepiness Scale |
| DSS | Driver Sleepiness Scale |
| GSQS | Groningen Sleep Quality Scale |
| MCTQ | Munich Chronotype Questionnaire |

## Appendix A. Review Results of Field Studies from Literature

**Table A1.** Field studies using scheduled light exposure for night-shift workers.

| Authors, Reference | Treatment Period | 0- Condition | Lighting Setup | E (lx) ‡, Spectrum, Tcp (K) | Protocol †† | | N | Phase Shift | Subj. Sleep | Obj. Sleep | Sub. Alertness | Cog. Performance | Others |
|---|---|---|---|---|---|---|---|---|---|---|---|---|---|
| *Boivin et al.* [44] | | E = 111 lx | light box, blue-blocking goggles | 2000 lx Full | | a | 15 | + * (Mel., CBT) | X (TST) | | | | |
| *Boivin et al.* [45] | | E = 111 lx | light box, blue-blocking goggles | 2000 lx Full | | a | 15 | + ** (Mel., CBT) | X (TST) | + * (TST) | | | |
| *James et al.* [46] | | E = 111 lx | light box, sunglasses | 2000 lx Full | | a | 11 | + ** (Cort.) | | | | | |
| *Boivin et al.* [47] | | | light box, sunglasses | 1350 lx Full | | a | 15 | X | X | | X | + * | + *** (UaMT6s) |
| *Boudreau et al.* [48] | | | light box, sunglasses | 1350 lx ‡‡ Full | | a | 15 | X | X | + * (WASO) + ** (HRV) | + * (VAS) | X | X (mood) |
| *Budnick et al.* [29] | | | ambient | 4000–8000 lx | | | 29 | X | X | | X | | |
| *Huang et al.* [49] | | 100–400 lx | light box & sunglasses | 7000–10,000 lx Full | | | 92 | | + *** (ISI) | | | | + *** (HADS) |
| *Karchani et al.* [50] | | 300 lx | ambient | 2500–3000 lx | | | 90 | | | | + *** | | |
| *Lowden et al.* [51] | | 300 lx | ambient | 2500 lx, full (5000 K) | | b | 18 | + * (Mel.) | | + * (TST) | + * (KSS) | | |
| *Motamedzadeh et al.* [52] | | 352 lx, Tcp = 2500–3000 K | ambient | 354 lx, full (6500 K) | | | 30 | + ** (Mel.) | | | + *** (KSS) | X | |
| | | | | 350 lx, full (17,000 K) | | | 30 | + ** (Mel.) | | | + *** (KSS) | + * | |
| *Sasseville et al.* [23] | | 130 lx | special fixture & blue-blocking goggles | 200 lx (500 nm, 66 μW/cm²) | | c | 4 | | | + * (TST,SL) | | + * | X(errors) |

**Table A2.** Field studies using scheduled light exposure for night-shift workers.

| Authors, Reference | Treatment Period | 0- Condition | Lighting Setup | E (lx) ‡, Spectrum, Tcp (K) | Protocol †† | | N | Results | | | | | |
|---|---|---|---|---|---|---|---|---|---|---|---|---|---|
| | | | | | | | | Phase Shift | Subj. Sleep | Obj. Sleep | Sub. Alertness | Cog. Performance | Others |
| Yoon et al. [25] | | 100–500 lx + 10,000 lx (1 h) | light box | 4000–6000 lx Full | | | 12 | | | + * (TST, SE) | X | + * | |
| | | | Light box & sunglasses | | | | 12 | | | + ** ( TST, SE) | X | + ** | |
| Bjorvatn et al, [3] | | | light box | 10,000 lx Full | | d,e | 17 | | x | | X (KSS) X (ATS) | | |
| | | | | | | d,f | 17 | | + * (SQ) | | + ** (KSS) + ** (ATS) | | |
| Bjorvatn et al. [53] | | 200–300 lx | light box | 10,000 lx, Full | | g | 17 | | + * (SL) | X | X (KSS) X (ATS) | | X (reaction time) |
| | | | | | | g | 17 | | + * (TST) | X | X (KSS) + * (ATS) | | X (reaction time) |
| Ross et al. [27] | | 300 lx (red light) | Light box | 2500–3000 lx Full | | | 14 | + ** | + * (SL) | | | | X (mood) |
| Tanaka et al. [28] | | Eh = 530–640 lx | light box | 6666 lx full | | b | 61 | | + * (VAS) | | + ** (KSS) + *** (CISQ) | X(PVT) | X(errors) |
| Thorne et al. [26] | | Sunglasses | Light box | 3000 lx ‡‡, full 1 mW/cm² | | h | 10 | X | + * (SQ) | + ** (TST) + * (SE) | | | |

† dark-grey areas = nightshift; light-grey areas = day shift; T = light exposure treatment; ‡ presumably measured at eye-level; ‡‡ continuously measured; †† black areas = light exposure; grey areas = work shift; dashed areas = varying timing; a as workload permitted; b self-selected; c progressive extension of exposure duration to 7 h across days; d start of treatment scheduled 3 h before habitual awakening; e delay of scheduled start 1 h per day; f delay of scheduled start 2 h per day; g start of treatment ≥ 2 h before habitual awakening; h advance of scheduled start 1 h per day; * = $p < 0.05$; ** = $p < 0.01$; *** = $p < 0.001$; + = positive effect; x = no effect; CBT = core body temperature; UaMT6s = urinary 6-sulfatoxymelatonin; Cort. = cortisol; ISI = Insomnia Severity Index; Mel. = melatonin; SE = sleep efficiency; SL = sleep latency; SQ = sleep quality; TST = total sleep time; HADS = hospital anxiety depression scale; WASO = wake after sleep onset; ATS = accumulated time of sleepiness; VAS = visual analogue scale; CISQ = checklist individual strength.

## Appendix B. Results from the Linear Mixed-Effects Models (LMM) Analyses

**Table A3.** Results of the LMM analyses of sleepiness (KSS).

| DV | Term | DF | F | p | $R^2$ | Post-hoc | $ICC_{Subject}$ |
|---|---|---|---|---|---|---|---|
| **During nightshift** | **Condition** | 1776.3 | 0.305 | 0.581 | 0 | | 0.393 |
| **During nightshift** | **Daytype** | 2776.6 | 6.34 | 0.002 ** | 0.016 | FN > N, LN; | |
| **During nightshift** | **Time of day** | 4776.1 | 51.53 | 0.001 *** | 0.202 | 6–8 h > 0–2 h; 6 h > 0–4 h | |
| **During nightshift** | **Condition × Daytype** | 2776.4 | 3.32 | 0.037 * | 0.007 | Plac: FN > LN <br> Plac: FN > Treat: N | |
| **During nightshift** | **Condition × Daytype × Time of day** | 8776.0 | 1.29 | 0.243 | 0.012 | FN, 4 h, Plac. > Treatm. <br> N, 8 h, Plac. > Treatm. | |
| **After nightshift** | **Condition** | 1416.2 | 0.968 | 0.326 | 0.036 | | 0.398 |
| **After nightshift** | **Daytype** | 2417.2 | 3.31 | 0.038 * | 0.018 | FN < N; | |
| **After nightshift** | **Time of Day** | 3417.0 | 3.44 | 0.017 * | 0.025 | 18 h > 20 h | |
| **After nightshift** | **Condition × Daytype** | 2415.9 | 0.191 | 0.827 | 0.002 | | |
| **D1, D2, R1, R2** | **Condition** | 1806.0 | 0.710 | 0.400 | 0 | | 0.178 |
| **D1, D2, R1, R2** | **Daytype** | 3805.6 | 2.68 | 0.046 * | 0.007 | D1 < D2, R1, R2; | |
| **D1, D2, R1, R2** | **Time of day** | 8805.8 | 13.03 | 0.001 *** | 0.117 | 12–20 h < 8–10 h, 22–24 h | |
| **D1, D2, R1, R2** | **Condition × Daytype** | 3804.8 | 0.856 | 0.462 | 0.005 | | |
| **D1, D2, R1, R2** | **Condition × Daytype × Time of day** | 15,804.5 | 0.677 | 0.809 | 0.014 | R1, 18 h, Plac. < Treatm. | |

Note. D1 = first day before nightshift, D2 = second day before nightshift, FN = first night of nightshift, N = nightshift, LN = last night of nightshift, R1 = first recovery day, and R2 = second recovery day. * = $p < 0.05$; ** = $p < 0.01$; *** = $p < 0.001$.

**Table A4.** Results of the LMM analyses of sleepiness parameters (DSS and KSS) after nightshift.

| DV | Term | DF | F | p | $R^2$ | Post-hoc | $ICC_{Subject}$ |
|---|---|---|---|---|---|---|---|
| **DSS** | **Condition** | 1130.3 | 2.10 | 0.149 | 0.016 | | 0.538 |
| **DSS** | **Daytype** | 2130.5 | 8.20 | <0.001 *** | 0.112 | FN, N > LN; | |
| **DSS** | **Condition × Daytype** | 2130.4 | 3.60 | 0.030 * | 0.052 | FN Plac > Treat: <br> Plac: LN < FN, N | |
| **KSS** | **Condition** | 1130.2 | 0.196 | 0.659 | | | 3.32 |
| **KSS** | **Daytype** | 2130.3 | 2.27 | 0.108 | | | |
| **KSS** | **Condition × Daytype** | 2130.2 | 1.59 | 0.208 | | | |

Note. D1 = first day before nightshift, D2 = second day before nightshift, FN = first night of nightshift, N = nightshift, LN = last night of nightshift, R1 = first recovery day, and R2 = second recovery day. * = $p < 0.05$; *** = $p < 0.001$.

**Table A5.** Results of the LMM analyses of sleep parameters bed time (BT), get-up time (GUT), time in bed (TIM), total sleep time (TST), sleep latency (SL), fragmentation index (FI), and Groningen Sleep Quality Index (GSQI).

| DV | Term | *DF* | *F* | *P* | $R^2$ | Post-hoc | $ICC_{Subject}$ |
|---|---|---|---|---|---|---|---|
| BT | Condition | 1226.4 | 0.09 | 0.766 | 0 | | 0 |
| BT | Daytype | 5226.5 | 3549.2 | <0.001 *** | 0.99 | D2, R1, R2 > FN, N, LN | |
| BT | Condition × Daytype | 5226.5 | 1.50 | 0.191 | 0.03 | D2: Plac. > Treat. | |
| GUT | Condition | 1230.1 | 2.15 | 0.144 | 0.01 | | 0.03 |
| GUT | Daytype | 5230.1 | 419.58 | <0.001 *** | 0.90 | D2, R1, R2 < FN, N, LN<br>FN, N > LN | |
| GUT | Condition × Daytype | 5230.1 | 1.93 | 0.09 | 0.04 | D2 > R2<br>R2: Treat. > Plac. | |
| TIB | Condition | 1229.7 | 0.033 | 0.857 | 0 | | 0.05 |
| TIB | Daytype | 5229.4 | 85.49 | <0.001 *** | 0.66 | D2, R1, R2 > FN, N, LN<br>FN, N > LN | |
| TIB | Condition × Daytype | 5229.5 | 2.35 | 0.042 * | 0.05 | R2: Treat. > Plac. | |
| TST | Condition | 1229.5 | 0.095 | 0.759 | 0 | | 0.05 |
| TST | Daytype | 5229.2 | 74.27 | <0.001 *** | 0.62 | D2 > FN, N, LN, R2<br>R1, R2 > N, LN<br>FN, N > LN | |
| TST | Condition × Daytype | 5229.3 | 2.65 | 0.024 * | 0.06 | R2: Treat. > Plac. | |
| SL | Condition | 1229.0 | 0.48 | 0.491 | 0 | | 0.22 |
| SL | Daytype | 5228.4 | 3.57 | 0.004 ** | 0.07 | R2 > FN, N | |
| SL | Condition × Daytype | 5228.4 | 2.91 | 0.014 * | 0.06 | D2: Treat. > Plac.<br>R2: Plac. > Treat | |
| SE | Condition | 1224.4 | 0.022 | 0.882 | 0 | | 0.49 |
| SE | Daytype | 5224.1 | 3.01 | 0.012 * | 0.06 | FN > LN, R2 | |
| SE | Condition × Daytype | 5224.1 | 1.92 | 0.092 | 0.04 | R2: Treat. > Plac. | |
| FI | Condition | 1227.9 | 1.35 | 0.246 | 0 | | 0.39 |
| FI | Daytype | 1227.9 | 2.16 | 0.059 | 0.05 | FN < R2 | |
| FI | Condition × Daytype | 5227.9 | 2.08 | 0.068 | 0.04 | | |
| GSQS | Condition | 1211.4 | 4.65 | 0.032 * | 0.02 | Treat. > Plac. | 0.31 |
| GSQS | Daytype | 5211.3 | 7.41 | <0.001 *** | 0.15 | D2, FN < N, LN | |
| GSQS | Condition × Daytype | 5211.7 | 2.03 | 0.075 | 0.05 | FN: Treat. > Plac.<br>R1: Treat. > Plac. | |

Note. D2 = day before nightshift, FN = first night of nightshift, N = nightshift, LN = last night of nightshift, R1 = first recovery day, and R2 = second recovery day. * = $p < 0.05$; ** = $p < 0.01$; *** = $p < 0.001$.

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
