# Peer review of "Can Special Light Glasses Reduce Sleepiness and Improve Sleep of Nightshift Workers? A Placebo-Controlled Explorative Field Study"

_2624-5175, doi:10.3390/clockssleep2020018_

Round 1
Reviewer 1 Report
This is one of the rare, well-controlled field studies on the effects of light in nightshift workers and the first one, I believe, with light goggles. I would like to express my appreciation for setting up and performing this study. Nevertheless, I do also have some comments on the paper, some major, some minor. Please see below.
Line 89: please add values for the treatment condition as well, staying stable is not necessarily better since there is a reduction in sleepiness in the control condition.
Figure 1: please explain the error bars
Line 112: Please explain in words what this means, worse sleep in treatment condition?
Line 117: please add: compared to what?
Line 132: I miss a discussion on the fact whether you do want to induce a phase shift and light at night in the context of long-term health.
Discussion in general: I sometimes get the feeling that the results are discussed in a more positive sense than the data indicate. e.g. in line 142, add "after the first night" because DSS is only higher in the placebo after the first night. In addition: line 149, you could also say that in the placebo condition, sleepiness dropped in the course of the period, while this was not the case in the treatment condition. The same holds for DSS. Should this be interpreted as a positive outcome for the treatment condition? Lines 153-154, this sounds more positive than it is and is contradictory to line 150-151.
Line 156: It is a bit confusing that there are several tables 1. In the main document, in the Appendix, and in the supplemental documentation. Please improve.
Line 157-166: The extra light treatment after waking up was somewhere in the afternoon. Based on a regular phase response curve, it seems very unlikely that this light exposure prevented a phase delay. In addition in lines 194-195, you discuss that the increased light exposure (measured in photopic lux) in the afternoon is marginal, so this seems not logical to me. I was rather thinking in the opposite way, that there is a larger phase delay in the treatment condition because of late evening light exposure and blue-blockers in the early morning. This may have shifted the drop in alertness to the early morning in the course of 3 or 5 night shifts. Please explain why you think that the opposite is true and discuss this in the discussion.
Line 177: a significant effect on D2 cannot be attributed to the treatment, because it is prior to the treatment.
Lines 191-194: Discussing illuminance as photopic lux in this study is not appropriate in my opinion. Melanopic lux EDI is the relevant factor and that will show a much bigger effect. Why not try to do that with the blue sensor of the Lightwatcher?
Lines 198-200: At night you doubled the amount of exposure to short wavelengths in the treatment condition, how does this compare to dose-response curves for alertness and phase delay? Do you expect an effect based on your light exposure? There are studies showing that intermittent light exposure can be highly effective in inducing a phase delay, please compare with your experiment of using 4 blocks of 15 minutes.
Line 217: isn't this in contrast to your own reasoning of prevention of a phase delay in lines 157-166?
Line 227: a description of the sensitivity of Light Watcher sensors is given in S1 not in figure 5
Line 236: if you think that season might have played a role, would it be an idea to test this by adding this factor to your LMM? By the way, your research period includes a shift to Daylight Saving Time, how did you deal with this in your data collection and analyses?
Line 250: What do you mean with a potential mistake in the administration of the light glasses? This does not sound very trustworthy.
Line 253: Methods..in a field study like this, I do expect missing data to occur. Was this the case, and if so how did you deal with this in your analyses? Please explain.
Line 269: where = were
Figure 5: how is it possible that there are more long wavelengths in the blue-blocking goggles than in the light glasses? You give absolute values on the y-axis. These wavelengths are not filtered out in the blue Led glasses are they, neither they are added by the blue blocking glasses? Please explain the difference in the long wavelengths between glasses.
I would advise using the new CIE S02618 toolbox to use the standard units for the five alpha opics.
lines 402-403: I do not understand this last conclusion, what are the arguments? Line 401 it is stated: "none of the sleep parameters seem to be impacted by the light glasses". Please explain.
Reviewer 2 Report
General comments: This is a thorough report on the efficacy of a novel, portable lighting intervention for night shift-workers. The topic is of extreme interest, as increasing evidence demonstrates that light interventions must be individually tailored, or "personalized". Additionally, the portability of the device is both novel and potentially ground-breaking in that individuals may receive therapy while performing their duties. The authors do a good job of describing their somewhat small and specific results in the context of the analyses. More information on how the devices may or may not have interfered with work, or other measures such as tolerability (were they uncomfortable? too bright?) would have been good to have as it informs the feasibilty of the use of such devices. Authors mention tracking of adherence/usage, but those data are not reported.It is a shame that phase markers were not used, though this is quite difficult to do in field studies and the authors do acknowledge this as a limitation. Additionally, it is mentioned in the discussion that this was done to reduce participant burden, which must be a consideration in these types of studies. The authors are applauded for conducting what must have been a difficult study and for investigating a novel and important form of treatment for shiftworkers.
Specific comments:
1. "This should generate a phase position compatible with both daytime and nighttime sleep" should be qualified: "This should generate a phase position at least partially compatible with both daytime and nighttime sleep", as it is not fully compatible with either. It also should be mentioned that refs 18-20 are all simulated night work, and not field studies. This concept has not been demonstrated in a workplace setting. 2. How often do they work switch from nights to days? The participants are described as "rotating" but no mention of the schedule is made. Some details can be gleaned from close inspection of the protocol figure but a short description would be helpful. 8 h shifts? 11p-7am? Additionally, some more detail of how many individuals worked X number of nights in a row, etc would be good to note. 3. In section 2.2, DSS is mentioned without being defined. 4. In Section 2.3 and Figure legend 3, GSQS results are mentioned without any indication of whether high scores are good or bad. It would be helpful to the reader to have some sense of that rather than having to go look for it in the back of the paper (for example PSQI, another measure of "sleep quality", high levels are bad, even though it is called a "sleep quality" scale rather than a "sleep disruption" scale). Additionally, in the discussion the authors state, "sleep quality (GSQG) in our study, was significantly lower under Treatment..." Suggest saying "significantly poorer" or something similar to clarify. 5. In the same two sections, it would be helpful to note to the reader that this is all watch-based outcomes. 6. I do not understand the n's for Table 1 - why are the n's different across condition? (n=22 in placebo [21 pos/1 neg] and n= 38 [33 pos/5 neg] in treatment). Is that a function of missing data for some participants in one condition but not another? So are the data a mixture of within-subjects and between subjects? If so, were LMM used? Also, all the items in Table 1 have an "a" after them (may just be a formatting issue in the making of the pdf for review?). 7. re: the single time point of more sleepiness in treatment condition at a single time point on the first recovery day: it seems like an alternative explanation might be that the workers entrained better in the treatment group and were therefore still in a nocturnal phase position on recovery day 1 (with higher sleepiness during the time when they were sleeping on previous days). So it might be that the light after the night shift was actually insufficient to advance them back to their diurnal phase. 8. The ocular exposure of ~ 15 lux is quite low. Was this value selected for a certain reason? (i.e., are there settings on the glasses, or is 15 lux the output and it cannot be adjusted). I might expect a little more discussion of why the authors suspected this low amount would be beneficial, especially after sleep on night shifts (during daylight hours). There is a brief mention of it being 20% more on the night shifts, which is more understandable given the very low levels (~80 lux). Also, there is a lot of emphasis on the potential of a phase-advancing effect from the light upon waking, which is inconsistent with the relatively weak stimulus. 9. The authors should probably include reference to this review or one like it - PVT and KSS have been examined alongside one another in many shiftwork studies: Åkerstedt, T., Anund, A., Axelsson, J., & Kecklund, G. (2014). Subjective sleepiness is a sensitive indicator of insufficient sleep and impaired waking function. Journal of sleep research, 23(3), 242-254. 10. There seems to be an incomplete sentence: "Also given the result that the healthcare workers indicated a lower alertness (KSS-score) on the first night of shift while the PVT was equally impaired on subsequent nights." This also seems to argue against the point that comes immediately beforehand. 11. I would mention in the discussion (line 215) that the two measures do in fact correlate in your sample. 12. Table 2 - MSFESC not defined 13. The protocol figure (Figure 6) is very difficult to read. In particular, the small letters inside the bars are difficult to discern and distinguish from one another. Some of this may be due to the pdf/proof but suggest larger font and/or bolding or some other way of clarifying. 14. I do not understand the statement: "These findings suggest that light glasses may help prevent after-effect of night-shift work on sleep." If the glasses made sleep quality worse, how is that preventing aftereffects? are the authors referring here to inhibiting sleep rebound? Please clarify. 15. The authors are applauded for included the spectrum of the blueblocking glasses, and for presenting all of the light data from the wearable. However, the authors should recognize the possibility that some of the effects are driven by the blue-blocking glasses. Indeed, effects were small and individuals rated both conditions as effective. Also, the light was not extremely bright. 16. What was the rationale for omitting naps? Ease of analyses? Night shift workers often sleep in more than one interval. 17. The authors may want to explicitly mention that adaptation is shown in terms of KSS in both conditions. This is not consistently demonstrated and often outcome-dependent, so it is probably worth drawing some attention to.Author Response
Please see the attachment
